# Efficient Message-Passing Transformer for Error Correcting Codes

**Seong-Joon Park**[1,*]  **Taewoo Park**[1]  **Hee-Youl Kwak**[2,†]
**Sang-Hyo Kim**[3]  **Yongjune Kim**[1,†]  **Jong-Seon No**[4]
[1]POSTECH, [2]University of Ulsan, [3]Sungkyunkwan University, [4]Seoul National University
`joonpark2247@gmail.com, parktaewoo@postech.ac.kr, ghy1228@gmail.com,`
`iamshkim@skku.edu, yongjune@postech.ac.kr, jsno@snu.ac.kr`

## Abstract

Error correcting codes (ECCs) are a fundamental technique for ensuring reliable communication over noisy channels. Recent advances in deep learning have enabled transformer-based decoders to achieve state-of-the-art performance on short codes; however, their computational complexity remains significantly higher than that of classical decoders due to the attention mechanism. To address this challenge, we propose *EfficientMPT*, an efficient message-passing transformer that significantly reduces computational complexity while preserving decoding performance. A key feature of EfficientMPT is the *Efficient Error Correcting (EEC) attention* mechanism, which replaces expensive matrix multiplications with lightweight vector-based element-wise operations. Unlike standard attention, EEC attention relies only on query-key interaction using *global query vector*, efficiently encode global contextual information for ECC decoding. Furthermore, EfficientMPT can serve as a foundation model, capable of decoding various code classes and long codes by fine-tuning. In particular, EfficientMPT achieves $85\%$ and $91\%$ of significant memory reduction and $47\%$ and $57\%$ of FLOPs reduction compared to ECCT for $(648, 540)$ and $(1056, 880)$ standard LDPC codes, respectively.

## 1 Introduction

In modern digital communication, reliable data transmission over noisy channels is a primary objective. A fundamental approach to achieving this objective is the careful design of error correcting codes (ECCs) to correct noisy errors caused by channel impairments. ECC decoders have benefited from advances in deep learning, achieving improvements over conventional decoding algorithms for various code classes. Among them, neural network-based decoders using the transformer architecture (Vaswani et al., 2017) have achieved state-of-the-art decoding performance for short-length codes, since the transformer is one of the most powerful neural network structures (Choukroun & Wolf, 2022; 2023; Park et al., 2023; Choukroun & Wolf, 2024a;b). The first transformer-based decoder, known as the Error Correction Code Transformer (ECCT) (Choukroun & Wolf, 2022), employs masked self attention module. Recently, a more efficient transformer-based decoder, the Cross-Attention Message-Passing Transformer (CrossMPT) (Park et al., 2025) has been introduced, utilizing a masked cross-attention module, which is more effective than self-attention for error correction.

However, transformer-based decoders for ECCs face significant challenges due to the high computational complexity of the attention module (Choukroun & Wolf, 2022; 2023; Park et al., 2023; Choukroun & Wolf, 2024a;b). Specifically, the attention module exhibits a quadratic complexity of $\mathcal{O}(n^2)$ (Vaswani et al., 2017; Chang et al., 2023), where $n$ denotes the number of tokens or the code length in transformer-based decoders. This excessive complexity increases memory usage and computational complexity restricts the practical application of transformer-based decoders. Designing an efficient attention module is critical for reducing the decoding complexity of transformer-based

---

*The source code is available at `https://github.com/iil-postech/efficientmpt`.
†Corresponding authors

ECC decoders and allowing longer codes to benefit from the advantages of transformer-based decoders.

In this work, we propose a transformer-based decoder called *Efficient Message-Passing Transformer (EfficientMPT)*. EfficientMPT significantly reduces the number of parameters, memory usage, and computational complexity (i.e., floating point operations (FLOPs)) while maintaining superior decoding performance. Notably, these reductions become more pronounced as the code length increases, enabling our model to effectively decode long codes.

EfficientMPT consists of two types of EfficientMPT blocks, which iteratively update magnitude and syndrome embeddings, similar to message-passing algorithms. A key feature of EfficientMPT block is the *Efficient Error-Correcting (EEC) attention*, which relies only on *query-key interaction* using the parity-check matrix (PCM). During the query-key interaction, a *global query vector* is generated to effectively capture global contextual information across all syndrome and magnitude elements. This global query vector aggregates information from all positions of the syndrome and magnitude embeddings, providing a condensed representation of their overall structure. Leveraging the global query vector, EEC attention replaces the costly matrix multiplications typically used in standard attention mechanisms. Instead, it employs efficient element-wise vector-based operations, significantly reducing memory usage and computational complexity. Combined with this approach, EEC attention integrates the PCM into the attention module, enabling the model to embed the code structure. EfficientMPT updates magnitude and syndrome embeddings by simply adding the embedding with the attention output without any other complicated operations. This integration facilitates the training efficiency of EfficientMPT.

Furthermore, EfficientMPT features a bit position-invariant and code length-invariant architecture, making it a foundation model for ECC decoding. EfficientMPT, trained across several codes simultaneously, achieves notable decoding performance on trained codes. For unseen codes, the fine-tuning technique enables EfficientMPT to decode without the need to train the decoder from scratch.

Our EfficientMPT significantly reduces i) GPU memory usage, ii) FLOPs, and iii) the number of parameters while showing the state-of-the-art decoding performance for transformer-based decoders. The complexity reduction scales favorably with increasing code length. We believe that this represents an important step toward enabling efficient decoding for a wide range of lengths with transformer-based decoders.

## 2 RELATED WORK

Transformer-based decoders are a subclass of model-free neural decoders. Model-free neural decoders adopt general neural network architectures–fully-connected network (Gruber et al., 2017; Kim et al., 2018), recurrent neural network (Bennatan et al., 2018))–for ECC decoding, without relying on existing decoding algorithms (Dai et al., 2021; Buchberger et al., 2021; Kwak et al., 2023). Model-free decoders using transformer architectures are referred to as transformer-based decoders.

The first transformer-based decoder, ECCT (Choukroun & Wolf, 2022), employs a masked self-attention module to enhance training efficiency and decoding performance. The mask matrix, derived from a PCM, is designed to embed the code structure and capture relationships between bit positions. Building on this approach, several transformer-based decoders have been proposed. An extension of ECCT with multiple masks (Park et al., 2023) was introduced to improve its decoding performance by increasing the diversity of the mask matrices. Furthermore, a foundation model for ECCT (FECCT) Choukroun & Wolf (2024a) was proposed, enabling the decoding of various code classes by training a single model. This approach demonstrates that transformer-based decoders can indeed function as foundation decoders. Also, an end-to-end learning framework for transformer-based decoding was proposed (Choukroun & Wolf, 2024b).

However, all these methods use the concatenation $\tilde{y}$, which combines the magnitude $|y|$ and syndrome $s(y)$ of the received vector $y$ (i.e., $\tilde{y} = [|y|, s(y)]$) for the masked self-attention module. This results in a large attention map with high computational complexity and memory usage. The attention map of the original ECCT (Choukroun & Wolf, 2022) is sparse, as the mask matrix discards unrelated positions entirely. However, the approaches in (Choukroun & Wolf, 2024a;b) introduce a dense matrix that is added element-wise to the attention map. Instead of enforcing sparsity by

masking out connections, this matrix applies a continuous weighting to the attention scores, thereby leading to dense attention maps.

In contrast, CrossMPT (Park et al., 2025) employs a masked cross-attention module and processes magnitude $|y|$ and syndrome $s(y)$ separately, as they exhibit distinct characteristics. CrossMPT iteratively updates their embeddings through two cross-attention modules, which improves decoding performance. This approach also reduces decoding complexity by decreasing the attention map size and increasing the sparsity of the mask matrix. Similarly, the proposed EEC attention module processes the magnitude and syndrome iteratively, with a further simplified attention mechanism. As a result, it significantly reduces the number of parameters, memory usage, and FLOPs, without compromising the decoding performance achieved by CrossMPT.

# 3 BACKGROUND

## 3.1 ERROR CORRECTING CODES

Consider a linear block code $C$, defined by a generator matrix $\mathbf{G} \in \mathbb{F}_2^{k \times n}$ and a PCM $\mathbf{H} \in \mathbb{F}_2^{(n-k) \times n}$, where they satisfy $\mathbf{G}\mathbf{H}^\top = 0$ over $\{0,1\}$ with modulo 2 addition. A codeword $x \in C \subseteq \{0,1\}^n$ is encoded by multiplying the message $m$ of size $k$ with $\mathbf{G}$ (i.e., $x = m\mathbf{G}$). Let $x_s$ be the binary phase shift keying (BPSK) modulated signal of $x$, where $x_s = 1 - 2x$, and let $y$ be the noisy channel output when $x_s$ is transmitted. In the additive white Gaussian noise (AWGN) channel, this can be modeled as $y = x_s + z$, where $z \sim \mathcal{N}(0, \sigma^2)$. The decoder's objective is to recover the original codeword $x$ by correcting errors caused by noise. Upon receiving $y$, the decoder computes the syndrome $s(y) = \mathbf{H}y_b$, where $y_b = \text{bin}(\text{sign}(y))$. Here, $\text{sign}(a)$ returns $+1$ if $a \geq 0$ and $-1$ otherwise, while $\text{bin}(-1) = 1$ and $\text{bin}(+1) = 0$.

## 3.2 TRANSFORMER-BASED DECODERS

The transformer-based decoders employ a syndrome-based preprocessing method to address the overfitting problem (Bennatan et al., 2018; Choukroun & Wolf, 2022). In this preprocessing step, decoders generate the magnitude vector $|y| = (|y_1|, \ldots, |y_n|)$ and the syndrome vector $s(y) = (s(y)_1, \ldots, s(y)_{n-k})$ from the received vector $y$, which are then used as input to the transformer. The magnitude and syndrome vectors are used to estimate the multiplicative noise $\tilde{z}_s$, defined as: $y = x_s + z = x_s\tilde{z}_s$. The goal is to estimate $\tilde{z}_s$ from the received vector $y$. The decoder function $f$ outputs $\hat{z}_s$, and the estimated codeword $\hat{x}$ is computed as $\hat{x} = \text{bin}(\text{sign}(yf(y)))$. If the multiplicative noise is accurately estimated, then $\text{sign}(\tilde{z}_s) = \text{sign}(\hat{z}_s)$ and $\text{sign}(\tilde{z}_s\hat{z}_s) = 1$.

All prior transformer-based decoders incorporate a mask matrix within the attention mechanism to effectively learn relationships among codeword bits (Choukroun & Wolf, 2022; Park et al., 2023; Choukroun & Wolf, 2024a;b; Park et al., 2025). The decoding performance depends on the choice of the mask matrix since it discards less important relationships and focuses on critical ones to facilitate learning. The mask matrix is derived from the PCM $\mathbf{H}$, which explicitly defines direct relationships between codeword bits based on parity check equations.

## 3.3 ATTENTION MODULE

The attention module can be described by three components: Query ($\mathbf{Q}$), key ($\mathbf{K}$), and value ($\mathbf{V}$). Let $\mathbf{X}, \mathbf{X}' \in \mathbb{R}^{n \times d}$ be the input embeddings, where $n$ is the input vector size and $d$ is the embedding dimension. The input embedding $\mathbf{X}$ is projected to the query using the trainable weight matrix $\mathbf{W_Q} \in \mathbb{R}^{d \times d}$ by $\mathbf{Q} = \mathbf{X}\mathbf{W_Q}$ and the input embedding $\mathbf{X}'$ is projected to the key and value using the trainable weight matrices $\mathbf{W_K}, \mathbf{W_V} \in \mathbb{R}^{d \times d}$ by $\mathbf{K} = \mathbf{X}'\mathbf{W_K}$, and $\mathbf{V} = \mathbf{X}'\mathbf{W_V}$. To enable multi-head attention, $\mathbf{Q}$, $\mathbf{K}$, and $\mathbf{V}$ are split into $h$ heads, where each head has a reduced dimensionality $d_h = d/h$ such that $\mathbf{Q} = [\mathbf{Q}^1, \cdots, \mathbf{Q}^h]$, $\mathbf{K} = [\mathbf{K}^1, \cdots, \mathbf{K}^h]$, $\mathbf{V} = [\mathbf{V}^1, \cdots, \mathbf{V}^h]$, where $[\cdot, \cdot]$ denotes concatenation. Finally, the attention output $\mathbf{Y}$ can be computed as: $\mathbf{Y} = [\text{Attn}(\mathbf{Q}^1, \mathbf{K}^1, \mathbf{V}^1), \cdots, \text{Attn}(\mathbf{Q}^h, \mathbf{K}^h, \mathbf{V}^h)] \mathbf{W_O}$, where $\text{Attn}(\mathbf{Q}, \mathbf{K}, \mathbf{V}) = \text{softmax}(\mathbf{Q}\mathbf{K}^\text{T}/\sqrt{d_h}) \mathbf{V}$ and $\mathbf{W_O} \in \mathbb{R}^{d \times d}$ denotes the output weight matrix. This mechanism is also known as scaled dot-product attention. If $\mathbf{X} = \mathbf{X}'$, the operation is referred to as self-attention; if $\mathbf{X} \neq \mathbf{X}'$, it is called cross-attention.

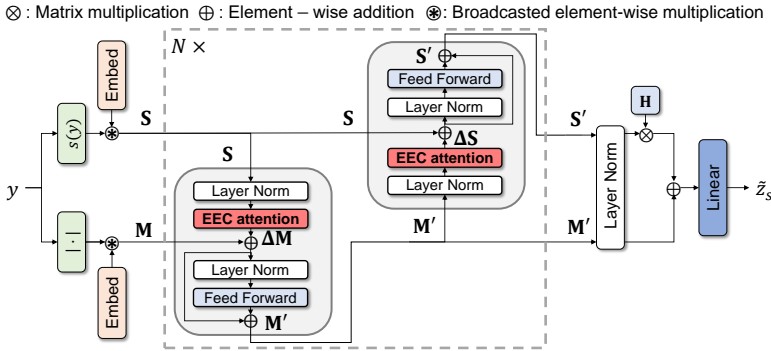

Figure 1: Architecture of EfficientMPT.

# 4 EFFICIENTMPT

In this section, we introduce a novel transformer-based decoder called EfficientMPT. Our proposed EfficientMPT (presented in Figure 1), incorporating EEC attention module (presented in Figure 2(c)), has the following three key distinctive characteristics.

**Query-key interaction with global context.** The EEC attention module focuses solely on query-key interaction with a global query vector. The global query vector encapsulates comprehensive contextual information from all syndrome or magnitude elements. This approach eliminates the use of the value vector $\mathbf{V}$, simplifying the process and reducing computational complexity.

**Efficient vector-based attention.** Standard attention mechanisms exhibit quadratic complexity $O(n^2)$ due to the matrix multiplication involved. In contrast, EEC attention generates a global query vector through row-wise summation and combines it with the key matrix using broadcasted element-wise multiplication. This design eliminates the quadratic matrix multiplication, resulting in linear-time complexity $O(n)$ for the attention operation. Consequently, this approach significantly reduces both memory usage and computational complexity.

**ECC specialized attention with PCM.** In the architecture of EfficientMPT, we incorporate the PCM to interchangeably convert between the magnitude and syndrome domains. This effectively embeds the code structure into the transformer, as the PCM defines the constraints that all valid code-words must satisfy. This approach is distinct from standard attention mechanisms, where the PCM is indirectly used to construct the mask matrix, whereas EfficientMPT utilizes the PCM directly. In other words, we develop a new attention mechanism specialized for ECC decoding.

**Efficient model for foundation ECC decoder.** The entire process in EfficientMPT is invariant to bit positions and code lengths, sharing the parameters across various code classes. Trained on multiple codes simultaneously, a single EfficientMPT model can achieve superior decoding performance across several code classes and even generalize to unseen codes with minimal fine-tuning.

## 4.1 ARCHITECTURE OF EFFICIENTMPT

Figure 1 presents the architecture of the proposed EfficientMPT. We first generate the magnitude embedding $\mathbf{M} = [\mathbf{M}_1; \cdots; \mathbf{M}_n] \in \mathbb{R}^{n \times d}$ where $\mathbf{M}_i = |y_i| W_\mathrm{M}$ for $i = 1, \ldots, n$. Similarly, we generate the syndrome embedding $\mathbf{S} = [\mathbf{S}_1; \cdots; \mathbf{S}_{n-k}] \in \mathbb{R}^{(n-k) \times d}$ where $\mathbf{S}_i = s(y)_i W_\mathrm{S}$ for $i = 1, \ldots, n - k$. Here, $W_\mathrm{M} \in \mathbb{R}^{1 \times d}$ and $W_\mathrm{S} \in \mathbb{R}^{1 \times d}$ are trainable parameters. To establish EfficientMPT as a position-invariant and length-invariant foundation model for ECC decoding, we use the shared $W_\mathrm{M}$ for magnitude embeddings and the shared $W_\mathrm{S}$ for syndrome embeddings (Choukroun & Wolf, 2024a).

Two gray boxed blocks in Figure 1 are referred to as EfficientMPT blocks. The first EfficientMPT block, located on the left, contains an ECC attention module that updates the magnitude embedding $\mathbf{M}$ to $\mathbf{M}'$ using the syndrome embedding $\mathbf{S}$. The query $\mathbf{Q}_1$ and key $\mathbf{K}_1$ are computed as follows:

$$\mathbf{Q}_1 = \mathbf{S} \mathbf{W_Q}, \quad \mathbf{K}_1 = \mathbf{S} \mathbf{W_K}$$

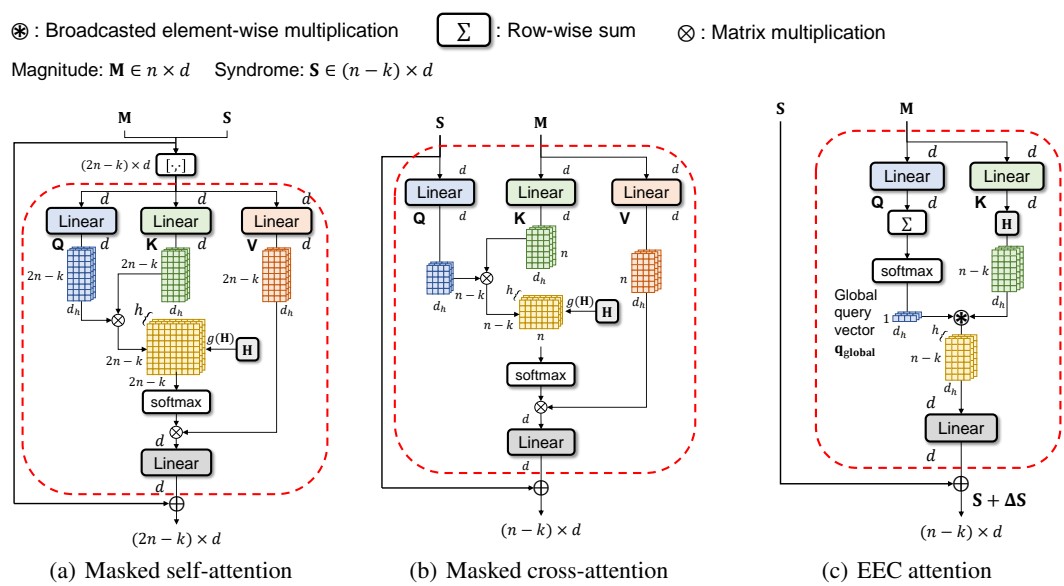

Figure 2: Comparison of attention modules for transformer-based ECC decoders. (a) Masked self-attention module (Choukroun & Wolf, 2022), (b) masked cross-attention module (Park et al., 2025), and (c) proposed EEC attention module.

By applying the proposed EEC attention, which will be explained in the next subsection, we obtain the attention output $\Delta \mathbf{M}$, which is added to the magnitude embedding $\mathbf{M}$. The result is then passed through a normalization layer, followed by a feedforward layer with a residual connection, to obtain the updated magnitude embedding $\mathbf{M}'$. Note that the query and key are derived from the same embedding type, which differs from the configuration used in CrossMPT.

The second EfficientMPT block updates the syndrome embedding using the updated magnitude embedding $\mathbf{M}'$. The query $\mathbf{Q}_2$ and key $\mathbf{K}_2$ are computed as follows:

$$\mathbf{Q}_2 = \mathbf{M}'\mathbf{W_Q}, \quad \mathbf{K}_2 = \mathbf{M}'\mathbf{W_K}.$$

Similar to updating the magnitude embedding, adding the attention output $\Delta \mathbf{S}$ to the syndrome embedding $\mathbf{S}$ efficiently produces the updated $\mathbf{S}'$, which is then utilized to refine the magnitude embedding in the next EfficientMPT block. This process is repeated $N$ times.

Finally, two output embeddings (magnitude and syndrome embeddings) from the last EfficientMPT block pass through a normalization layer. The magnitude embedding is then added to the resized syndrome embedding, which is resized from $(n-k) \times d$ to $n \times d$ by multiplying the transpose of the binary PCM $\mathbf{H}$. The combined output embedding then passes through a fully connected layer which reduces the $n \times d$ embedding to a one-dimensional $n$ vector. We want to note that all trainable parameters in EfficientMPT are position-invariant and length-invariant, which indicates that EfficientMPT can be utilized as a foundation decoder for ECCs.

## 4.2 EFFICIENT ERROR CORRECTING (EEC) ATTENTION

Figure 2 illustrates three different attention modules employed in transformer-based decoders. As shown in Figure 2(a), the masked self-attention module used in (Choukroun & Wolf, 2022; Park et al., 2023) concatenates magnitude and syndrome embeddings as input, resulting in a size of $(2n - k) \times d$. Thus, the attention-map has a size of $(2n - k) \times (2n - k)$ for each header. Figure 2(b) shows the masked cross-attention module employed in CrossMPT (Park et al., 2025) for updating the syndrome embedding. As previously noted, CrossMPT utilizes two distinct masked cross-attention modules to separately update magnitude and syndrome embeddings. Each cross-attention has an attention map of size $(n - k) \times n$. Since CrossMPT employs two cross-attention modules, the total size of the attention maps is $2n(n - k)$.

Figure 2(c) shows the proposed ECC attention module used to update the syndrome embedding, which is located in the second EfficientMPT block in Figure 1. Unlike standard attention mechanisms that rely on computationally expensive matrix multiplications, the ECC attention module employs row-wise summation and broadcasted element-wise multiplication as shown in Figure 3. This design choice is a key factor in reducing computational complexity.

Without loss of generality, we describe the EEC attention module that updates the syndrome embedding using information from the magnitude embedding. For the other attention module that updates the magnitude embedding, the roles are simply reversed, with the syndrome embedding used to update the magnitude embedding. The magnitude embedding $\mathbf{M} \in \mathbb{R}^{n \times d}$ is projected to the query $\mathbf{Q}$, key $\mathbf{K}$ by weight matrices $\mathbf{W_Q}, \mathbf{W_K} \in \mathbb{R}^{d \times d}$, respectively. Next, $\mathbf{Q}$ and $\mathbf{K}$ are split into $h$ attention heads, generating $\mathbf{Q}^i, \mathbf{K}^i \in \mathbb{R}^{n \times d_h}$ for $i = 1, \ldots, h$.

For the query, we compute the *global query vector* as follows:

$$\mathbf{q}_{\text{global}}^i = \text{softmax}\left(\sum_{j=1}^{n} \mathbf{Q}^i(j)\right) \in \mathbb{R}^{1 \times d_h},$$

where $\mathbf{Q}^i(j)$ denotes the $j$th row vector of $\mathbf{Q}^i$ (i.e., $\mathbf{Q}^i = [\mathbf{Q}^i(1); \cdots; \mathbf{Q}^i(n)]$). This global query vector is a critical component of the EEC attention mechanism, as it efficiently captures the global contextual information of the magnitude domain. By condensing the magnitude information across all elements into a single vector, the global query vector acts as a high-level representation that can be applied uniformly across the syndrome domain, allowing the model to propagate magnitude information in a highly efficient manner.

The next step is to project the magnitude information into the syndrome domain using the PCM $\mathbf{H}$. Specifically, the key matrix $\mathbf{K}^i \in \mathbb{R}^{n \times d_h}$ is transformed into $\mathbf{K_H}^i = \mathbf{H}\mathbf{K}^i \in \mathbb{R}^{(n-k) \times d_h}$. This transformation is essential because the PCM $\mathbf{H}$ inherently encodes the code structure, representing the relationships between the magnitude and syndrome elements. By applying $\mathbf{H}$, the model effectively maps the magnitude information into a representation within the syndrome domain. This allows the global query vector, which contains magnitude information, to be distributed across a matrix in the syndrome space, enhancing its effectiveness for error correction.

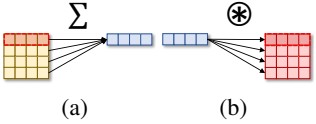

Figure 3: (a) Row-wise summation and (b) broadcasted element-wise multiplication.

The global query vector $\mathbf{q}_{\text{global}}^i$ is then broadcasted and element-wise multiplied with $\mathbf{K_H}^i$:

$$\Delta\mathbf{S} = \left[\mathbf{q}_{\text{global}}^1 \circledast \mathbf{K_H}^1, \cdots, \mathbf{q}_{\text{global}}^h \circledast \mathbf{K_H}^h\right] \mathbf{W_O} \in \mathbb{R}^{(n-k) \times d},$$

where $\circledast$ denotes the broadcasted element-wise multiplication. This operation allows the global context information of the magnitude, captured in the global query vector, to be propagated across all rows of the matrix $\mathbf{K_H}^i$ in the syndrome domain. Essentially, the global query vector broadcasts this global context information directly into the syndrome space, enabling efficient context sharing.

The output of EEC attention $\Delta\mathbf{S}$ represents the update for the syndrome embedding, learned from the magnitude embedding. This update is applied to the original syndrome embedding through simple addition: $\mathbf{S} \leftarrow \mathbf{S} + \Delta\mathbf{S}$. Then, the updated syndrome embedding pass through the normalization layer and the fully-connected layer to obtain the updated syndrome embedding. For the second attention module, the same process is applied to the magnitude embedding.

## 5 EXPERIMENTAL RESULTS

We adopt the same training setup as ECCT and CrossMPT: 1000 epochs, 1000 minibatches per epoch, and 128 samples per minibatch, using the Adam optimizer (Kingma & Ba, 2014). The learning rate is initialized at $10^{-4}$ and gradually decreases to $5 \times 10^{-7}$, using a cosine decay scheduler. The model is trained on all-zero codewords over $E_{\text{b}}/N_0$ from 3 dB to 7 dB, and tested on randomly generated codewords. All simulations are performed on NVIDIA GeForce RTX A5000 GPUs and AMD EPYC 7763 CPU.

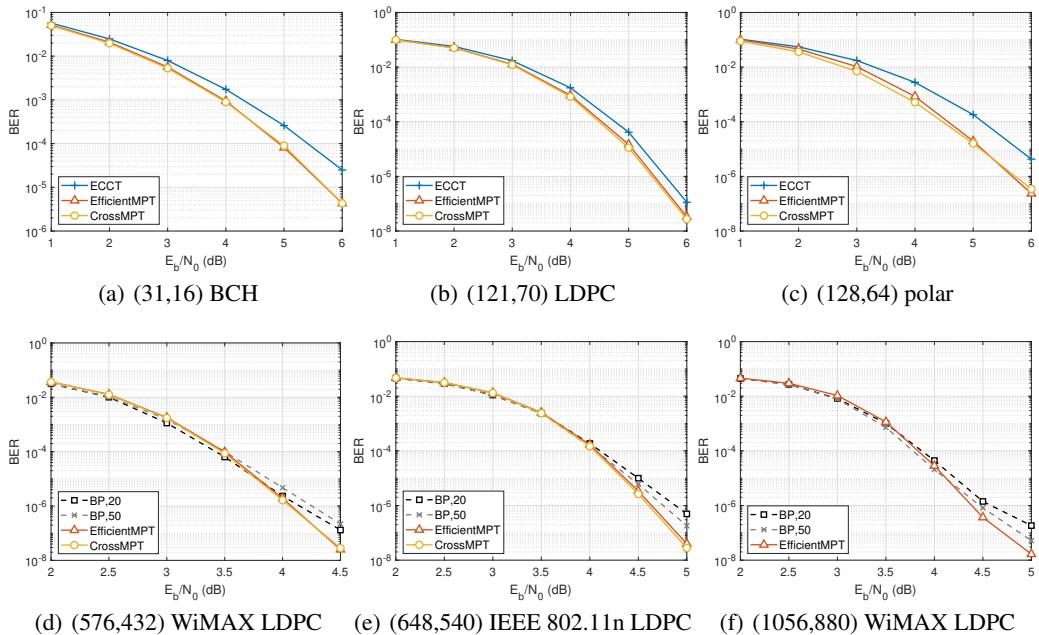

Figure 4: BER comparison for different code classes.

## 5.1 DECODING PERFORMANCE

Figures 4(a), 4(b), and 4(c) compare the BER performance of EfficientMPT, CrossMPT, and ECCT for short codes. All simulations are conducted with $h = 8$, $N = 6$, and $d = 128$. Across different code classes, EfficientMPT outperforms the original ECCT and achieves decoding performance comparable to CrossMPT. Additional results for various code parameters are provided in Appendix A.

Figures 4(d), 4(e), and 4(f) compare the BER performance of three standard LDPC codes: $(576, 432)$ WiMAX, $(648, 540)$ IEEE 802.11n, and $(1056, 880)$ WiMAX LDPC codes. All simulations are performed with $h = 8$, $N = 10$, and $d = 128$. Additionally, we include the performance of the belief propagation (BP) decoder. Both CrossMPT and EfficientMPT outperform the BP decoder with maximum iterations of 20 and 50. We note that ECCT for all these long codes and CrossMPT for $(1056, 880)$ LDPC codes could not be trained due to memory limitations in our simulation environment. However, even for lengths exceeding 1000, EfficientMPT remains trainable and achieves performance gains over the BP decoder. More results are presented in Appendix A, B, C, and D.

## 5.2 COMPLEXITY ANALYSIS

Table 1 compares GPU memory usage, computational complexity (FLOPs), and the number of trainable parameters across EfficientMPT, CrossMPT, and ECCT for various code classes. All results are obtained under the condition $N = 6$ and $d = 128$. For GPU memory usage, we measure the peak memory usage during the training of a single batch.

The table clearly demonstrates the GPU memory efficiency of EfficientMPT, which consistently requires less GPU memory across all code types. As the code length increases, the memory usage gap between EfficientMPT and other methods becomes more pronounced. For instance, for the (3328,640) 5G NR LDPC code, EfficientMPT requires only $0.35$ GB, whereas CrossMPT and ECCT require $8.42$ GB and $17.98$ GB, respectively. In other words, CrossMPT and ECCT require nearly $20\times$ and $50\times$ more memory than EfficientMPT. This substantial improvement is attributed to the proposed EEC attention module in EfficientMPT, which performs vector-based element-wise operations instead of matrix multiplications on large attention maps, such as $(2n - k)^2$ for ECCT and $2n(n - k)$ for CrossMPT.

Table 1: Comparison of GPU memory usage, FLOPs, and the number of parameters between EfficientMPT, CrossMPT, and ECCT.

| Codes | Parameter | Memory usage | | | FLOPs | | | # of parameters | | |
|---|---|---|---|---|---|---|---|---|---|---|
| | | EfficientMPT | CrossMPT | ECCT | EfficientMPT | CrossMPT | ECCT | EfficientMPT | CrossMPT | ECCT |
| WiMAX LDPC | $(576, 432)$ | **0.05 GB** **(16%)** | 0.13 GB (42%) | 0.31 GB (100%) | **0.92 G** **(56%)** | 1.11 G (67%) | 1.65 G (100%) | **1.09 M** **(64%)** | 1.70 M (100%) | |
| 802.11n LDPC | $(648, 540)$ | **0.05 GB** **(15%)** | 0.13 GB (38%) | 0.34 GB (100%) | **0.94 G** **(53%)** | 1.11 G (62%) | 1.78 G (100%) | **1.09 M** **(61%)** | 1.78 M (100%) | |
| WiMAX LDPC | $(1056, 880)$ | **0.07 GB** **(9%)** | 0.26 GB (32%) | 0.82 GB (100%) | **1.65 G** **(43%)** | 2.04 G (54%) | 3.80 G (100%) | **1.09 M** **(41%)** | 2.65 M (100%) | |
| WiMAX LDPC | $(2304, 1152)$ | **0.18 GB** **(3%)** | 2.63 GB (44%) | 6.02 GB (100%) | **8.18 G** **(36%)** | 12.27 G (55%) | 22.46 G (100%) | **1.09 M** **(11%)** | 9.60 M (100%) | |
| 5G NR LDPC | $(3328, 640)$ | **0.31 GB** **(2%)** | 8.42 GB (47%) | 17.98 GB (100%) | **21.44 G** **(34%)** | 34.65 G (55%) | 62.76 G (100%) | **1.09 M** **(5%)** | 21.98 M (100%) | |
| BCH | $(31, 16)$ | **36.19 MB** **(94%)** | 38.27 MB (99%) | 38.57 MB (100%) | **50.9 M** **(88%)** | 56.1 M (97%) | 57.9 M (100%) | **1.09 M** **(91%)** | 1.20 M (100%) | |
| BCH | $(63, 45)$ | **36.19 MB** **(92%)** | 38.54 MB (98%) | 39.52 MB (100%) | **90.2 M** **(85%)** | 99.9 M (94%) | 106.4 M (100%) | **1.09 M** **(90%)** | 1.21 M (100%) | |
| Polar | $(64, 32)$ | **36.19 MB** **(90%)** | 38.76 MB (97%) | 40.05 MB (100%) | **0.11 G** **(85%)** | 0.12 G (92%) | 0.13 G (100%) | **1.09 M** **(90%)** | 1.21 M (100%) | |
| Polar | $(128, 64)$ | **36.21 MB** **(76%)** | 40.44 MB (85%) | 47.62 MB (100%) | **0.22 G** **(79%)** | 0.25 G (89%) | 0.28 G (100%) | **1.09 M** **(90%)** | 1.21 M (100%) | |
| LDPC | $(121, 70)$ | **36.21 MB** **(82%)** | 40.07 MB (90%) | 44.43 MB (100%) | **0.20 G** **(77%)** | 0.23 G (88%) | 0.26 G (100%) | **1.09 M** **(89%)** | 1.23 M (100%) | |

Furthermore, EfficientMPT effectively reduces FLOPs compared to other methods, with the reduction becoming more pronounced as the code length increases. For the $(3328, 640)$ 5G NR LDPC code, EfficientMPT requires only 21.48 G, whereas CrossMPT and ECCT require 34.65 G and 62.76 G, respectively–representing a 38% reduction.

To further investigate the effectiveness of EfficientMPT, we additionally compare FLOPs in a graph, supplementing the results in Table 1. Figure 5 presents the FLOPs of 5G NR LDPC codes based on a base graph of size $10 \times 52$ with various lifting factors $Z = 1, 2, 4, 8, 16, 32, 64$ (Richardson & Kudekar, 2018). The lifting process generates LDPC codes with code length

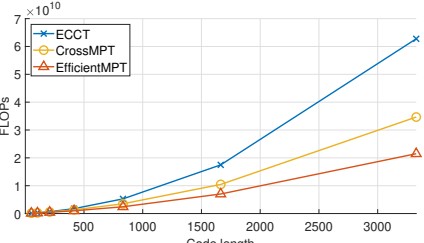

Figure 5: Comparison of FLOPs for EfficientMPT, CrossMPT, and ECCT for various 5G NR LDPC codes.

$n = 52 \times Z$ and dimension $k = 10 \times Z$. Since all 5G LDPC codes in the figure are derived from the same base graph, they share the same code structure regardless of the lifting factor. This setup ensures a fair comparison, focusing on the impact of code length while preserving the code structure. As shown in the figure, EfficientMPT exhibits nearly linear FLOPs, unlike other methods. This is because the only operation with quadratic complexity in Figure 2(c) is the multiplication with PCM, which is not a dominant computation compared to other operations.

The last column in Table 1 presents the number of trainable parameters. Note that CrossMPT and ECCT have the same number of parameters. Since all parameters in EfficientMPT are code-invariant, the number of parameters remains constant across different code classes as long as $N$ and $d$ are the same. For $N = 6$ and $d = 128$, EfficientMPT consistently maintains 1.09 M parameters (specifically, 1,097,649), whereas the number of parameters in CrossMPT and ECCT grows rapidly with increasing code length. These results demonstrate that EfficientMPT achieves superior efficiency in all aspects of computational complexity compared to CrossMPT and ECCT.

## 5.3 FOUNDATION EFFICIENTMPT

To train EfficientMPT as a foundation ECC decoder, we train EfficientMPT on four different codes: $(64, 32)$ LDPC, $(121, 60)$ LDPC, $(121, 70)$ LDPC, and $(121, 80)$ LDPC codes. We refer to this model as foundation EfficientMPT (FEfficientMPT). In Figure 6, we compare FEfficientMPT with EfficientMPT, ECCT, and foundation ECCT (FECCT) (Choukroun & Wolf, 2024a). The decoders trained on a single code (EfficientMPT, ECCT) are trained for 1000 epochs, whereas the foundation models (FEfficientMPT, FECCT) are trained for 4000 epochs since they are trained on four differ-

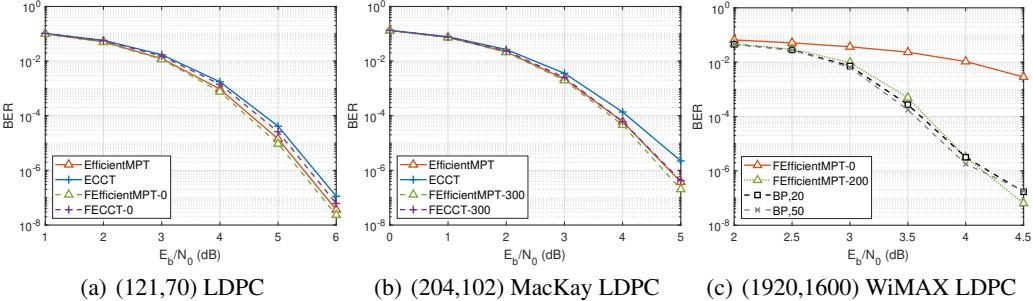

(a) (121,70) LDPC     (b) (204,102) MacKay LDPC     (c) (1920,1600) WiMAX LDPC

Figure 6: Decoding performance of (a) (121,70) LDPC code, (b) (204,102) MacKay LDPC code, and (c) (1920,1600) WiMAX LDPC code.

ent codes. Figure 6(a) shows the performance for the $(121, 70)$ LDPC code, which is one of the *trained* codes, while Figures 6(b) and 6(c) present the performance for the $(204, 102)$ MacKay and $(1920, 1600)$ WiMAX LDPC codes, which are *unseen* codes. The number following FEfficientMPT and FECCT denotes the number of fine-tuning epochs. For example, 'FEfficientMPT-0' denotes the foundation EfficientMPT without fine-tuning and 'FEfficientMPT-300' denotes the foundation EfficientMPT fine-tuned for 300 epochs.

In Figure 6(a), EfficientMPT and FEfficientMPT exhibit nearly identical performance, which means that the EfficientMPT model can be trained as a foundation ECC decoder without performance degradation. Figure 6(b) demonstrates that FEfficientMPT with fine-tuning (FEfficientMPT-300) outperforms ECCT and achieves decoding performance comparable to EfficientMPT, even for unseen codes. Notably, Figure 6(c) shows that although FEfficientMPT initially struggles to decode unseen codes, its performance steadily improves with fine-tuning and eventually surpasses the conventional BP decoder for the long WiMAX LDPC code. This result is practically meaningful because it demonstrates that FEfficientMPT model can be adapted for longer codes using a pretrained foundation model trained on short codes, eliminating the need for expensive full-scale training of long codes from scratch.

## 5.4 TRAINING H MATRIX

In the EfficientMPT architecture, we utilize (or multiply) the PCM to embed the code structure. For further analysis, we replace the PCM **H** with a trainable matrix. The trainable matrix is randomly initialized and optimized along with other parameters during training. Figure 7 shows the trainable matrix before and after the training on the $(64, 32)$ LDPC code. The darker the tone of each element, the larger its value. Initially, as shown in Figure 7(a), the trainable matrix is a random matrix. However, after training, Figure 7(b) reveals that the trained

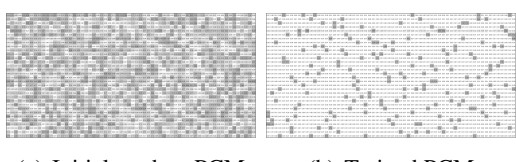

(a) Initial random PCM     (b) Trained PCM

Figure 7: The values of trainable matrix (a) before the training and (b) after the training.

matrix develops a structure closely resembling the original PCM. In Appendix F, we illustrate a comparison between the trained PCM and the original PCM. This observation further supports the use of the PCM in EfficientMPT.

## 6 CONCLUSION

In this work, we proposed EfficientMPT, a transformer-based decoder that overcomes the computational limitations of standard attention mechanisms in ECC decoding. EEC attention simplifies the standard attention by employing broadcasted element-wise operations and effectively learns the code structure by incorporating the PCM into the attention mechanism. Leveraging the proposed

EEC attention, we develop EfficientMPT, which significantly reduces the computational complexity while maintaining the superior decoding performance of CrossMPT. In addition, EfficientMPT can also serve as a universal and foundation-level decoder. A single model can achieve notable decoding performance across several code classes and even on unseen codes. The reduced computational complexity of EfficientMPT enables support for longer code lengths, allowing more practical codes to benefit from the advantages of transformer-based decoders and enhancing the feasibility of their application to long codes.

ACKNOWLEDGMENTS

This work was supported by Institute of Information & Communications Technology Planning & Evaluation (IITP) grant funded by the Korean Government (MSIT) (RS-2024-00398449, Network Research Center: Advanced Channel Coding and Channel Estimation Technologies for Wireless Communication Evolution) and IITP grant funded by the Korean Government (MSIT) (RS-2023-00229849, MPU/Connectivity/TinyML SoC Solution for IoT Intelligence with Foundry-Based eFLASH).

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

## A   ADDITIONAL RESULTS FOR VARIOUS CODE PARAMETERS

Table 2 demonstrates the decoding performance for various code classes and parameters for $h = 8$, $N = 6$, and $d = 128$. Across different code classes, EfficientMPT outperforms the original ECCT and achieves decoding performance comparable to CrossMPT.

Table 2: Comparison of the BER performance at three different $E_b/N_0$ values (4 dB, 5 dB, 6 dB) for EfficientMPT, CrossMPT, and ECCT (Choukroun & Wolf, 2022).

| Codes | Method | EfficientMPT | | | CrossMPT | | | ECCT | | |
|---|---|---|---|---|---|---|---|---|---|---|
| | | 4 dB | 5 dB | 6 dB | 4 dB | 5 dB | 6 dB | 4 dB | 5 dB | 6 dB |
| BCH | (31,16) | 9.11e-4 | 8.44e-5 | 3.58e-6 | 9.26e-4 | 9.63e-5 | 3.79e-6 | 1.68e-3 | 2.51e-4 | 2.35e-5 |
| | (63,36) | 7.44e-3 | 1.31e-3 | 9.85e-5 | 6.53e-3 | 9.98e-4 | 8.49e-5 | 7.75e-3 | 1.29e-3 | 1.12e-4 |
| | (63,45) | 3.26e-3 | 3.20e-4 | 1.50e-5 | 2.74e-3 | 2.74e-4 | 9.10e-6 | 3.70e-3 | 4.14e-4 | 1.79e-5 |
| Polar | (64,32) | 5.86e-4 | 4.10e-5 | 1.56e-6 | 5.51e-4 | 4.70e-5 | 1.66e-6 | 9.21e-4 | 7.95e-5 | 4.46e-6 |
| | (64,48) | 1.73e-3 | 1.88e-4 | 1.63e-5 | 1.49e-3 | 1.67e-4 | 1.22e-5 | 1.73e-3 | 2.12e-4 | 1.53e-5 |
| | (128,64) | 8.68e-4 | 2.27e-5 | 3.25e-7 | 5.40e-4 | 1.35e-5 | 3.88e-7 | 2.69e-3 | 1.77e-4 | 5.13e-6 |
| LDPC | (121,60) | 3.63e-3 | 1.18e-4 | 3.99e-7 | 3.23e-3 | 9.47e-5 | 3.83e-7 | 5.68e-3 | 2.46e-4 | 1.67e-6 |
| | (121,70) | 9.50e-4 | 1.23e-5 | 4.26e-8 | 8.59e-4 | 1.13e-5 | 2.46e-8 | 1.66e-3 | 3.68e-5 | 1.10e-7 |
| | (121,80) | 3.51e-4 | 4.11e-6 | 1.68e-8 | 3.38e-4 | 2.89e-6 | 1.31e-8 | 6.5e-4 | 1.0e-5 | 7.25e-8 |

## B   COMPARISON WITH DC-ECCT AND E2E DC-ECCT

Table 3: BER comparison between DC-ECCT, E2E ECCT, and EfficientMPT

| Method | DC-ECCT | | | E2E ECCT | | | EfficientMPT | | |
|---|---|---|---|---|---|---|---|---|---|
| Parameter | 4 dB | 5 dB | 6 dB | 4 dB | 5 dB | 6 dB | 4 dB | 5 dB | 6 dB |
| (31,16) BCH | 8.50e-4 | 6.19e-5 | 3.58e-6 | 7.54e-4 | 1.14e-4 | 2.65e-6 | 7.13e-4 | 5.49e-5 | 2.20e-6 |
| (64,32) Polar | 5.81e-4 | 2.78e-5 | 1.08e-6 | 5.05e-4 | 3.10e-5 | 2.07e-6 | 4.33e-4 | 2.83e-5 | 1.02e-6 |
| (49,24) LDPC | 3.81e-3 | 1.43e-3 | 7.39e-4 | 1.87e-3 | 1.55e-4 | 4.42e-6 | 1.14e-3 | 6.15e-5 | 1.14e-6 |

Table 3 compares the BER performance of EfficientMPT with DC-ECCT, E2E DC-ECCT (Choukroun & Wolf, 2024b), which are improved architectures of ECCT. All results are obtained by training with 1024 samples per minibatch, for 1000 epochs, with 1000 minibatches per epoch. The decoding performances of DC-ECCT and E2E DC-ECCT are obtained from Choukroun & Wolf (2024b). For all BCH code, polar code, and LDPC code, EfficientMPT outperforms all DC-ECCT and E2E DC-ECCT.

## C   COMPARISON WITH SUCCESSIVE CANCELLATION LIST DECODER

We compare the decoding performance of the successive cancellation list (SCL) decoder with ECCT, CrossMPT, and EfficientMPT for polar codes. The performance of the SCL decoder is taken from

Table 4: Comparison with SCL decoder for polar codes

| Method | SCL(L=1) | | | SCL(L=4) | | | ECCT | | | CrossMPT | | | EfficientMPT | | |
|---|---|---|---|---|---|---|---|---|---|---|---|---|---|---|---|
| Parameter | 4 dB | 5 dB | 6 dB | 4 dB | 5 dB | 6 dB | 4 dB | 5 dB | 6 dB | 4 dB | 5 dB | 6 dB | 4 dB | 5 dB | 6 dB |
| (64,32) | 6.76e-4 | 6.31e-5 | 1.89e-6 | 3.01e-4 | 2.25e-5 | 7.99e-7 | 9.21e-4 | 7.95e-5 | 4.46e-6 | 5.53e-4 | 4.68e-5 | 1.66e-6 | 5.86e-4 | 4.10e-5 | 1.56e-6 |
| (64,48) | 2.05e-3 | 2.23e-4 | 1.72e-5 | 1.24e-3 | 1.79e-4 | 1.31e-5 | 1.73e-3 | 2.09.E-37 | 1.53e-5 | 1.40e-3 | 1.67e-4 | 1.22e-5 | 1.73e-3 | 1.88e-4 | 1.63e-5 |
| (128,64) | 2.32e-4 | 8.38e-6 | 1.12e-6 | 6.77e-5 | 1.93e-6 | 2.72e-8 | 2.69e-3 | 1.77e-4 | 5.13e-6 | 5.42e-4 | 1.35e-5 | 3.89e-7 | 8.68e-4 | 2.27e-5 | 3.25e-7 |

Choukroun & Wolf (2022) and Park et al. (2025). Since the SCL decoder for polar codes has been extensively studied over time, while transformer-based decoders are still in the early stages of research, it remains challenging for transformer-based methods to outperform the SCL decoder. Nevertheless, both CrossMPT and EfficientMPT demonstrate significantly better performance than the original ECCT for polar codes, and achieve comparable performance to the SCL decoder for the $(64, 48)$ polar code. In other words, we can improve decoding performance over the original transformer-based decoder while greatly reducing decoding complexity. These findings suggest that continued research in this direction can lead to further meaningful advancements in transformer-based ECC decoding.

# D    DECODING RESULTS FOR LONGER CODES

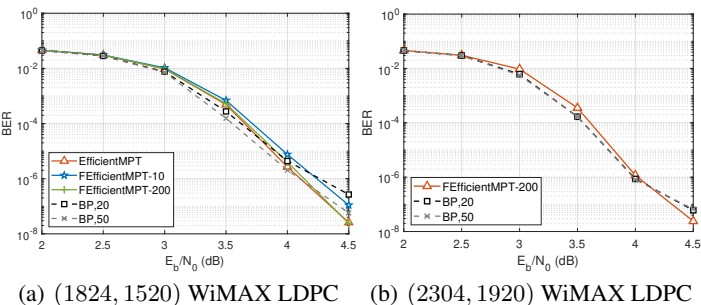

(a) $(1824, 1520)$ WiMAX LDPC         (b) $(2304, 1920)$ WiMAX LDPC

Figure 8: Decoding performance of (a) $(1824, 1520)$ WiMAX LDPC code and (b) $(2304, 1920)$ WiMAX LDPC code for the proposed method and BP decoders

Figure 8 compares the decoding performance of the proposed EfficientMPT with that of conventional belief propagation (BP) decoders for long codes. For the $(1824, 1520)$ WiMAX LDPC code, EfficientMPT is trained from scratch and FEfficientMPT is fine-tuned with the pre-trained EfficientMPT model. The results demonstrate that with only 10 epochs of fine-tuning (FEfficientMPT-10), the model already achieves performance comparable to the BP decoder. Furthermore, when fine-tuned for 200 epochs (FEfficientMPT-200), the performance closely matches that of the EfficientMPT trained from scratch and surpasses the BP decoder. In addition, FEfficientMPT with fine-tuning also outperforms the BP decoder with maximum iterations of 20 and 50 for the $(2304, 1920)$ WiMAX LDPC code.

This range of code lengths is particularly challenging for conventional transformer-based decoders. EfficientMPT, however, enables a broader range of code lengths to benefit from the advantages of transformer-based decoding. Rather than relying on code-specific models, a single EfficientMPT model achieves strong decoding performance across various code types, demonstrating the universality of foundation decoders. When deployed as a foundation decoder, such a model can significantly reduce hardware complexity and power consumption by supporting multiple generations of codes within a unified architecture—effectively functioning as a "multiple-in-one" decoder. The effectiveness of EfficientMPT also extends the applicability of transformer-based decoders to a wider range of practical scenarios.

# E FEFFICIENTMPT WITH VARIOUS FINE-TUNING SETTINGS FOR UNSEEN CODES

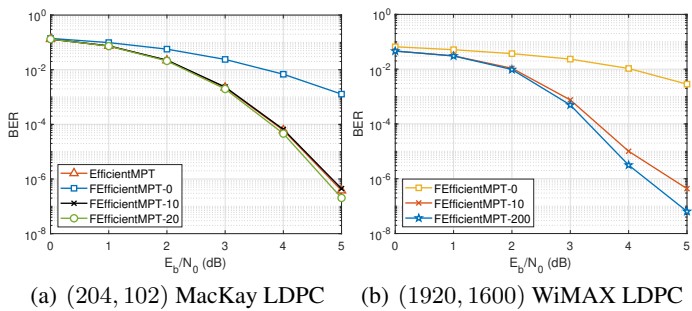

(a) $(204, 102)$ MacKay LDPC      (b) $(1920, 1600)$ WiMAX LDPC

Figure 9: Decoding performance of (a) (204,70) MacKay LDPC code and (b) (1920,1600) WiMAX LDPC code for various fine-tuning epochs.

Figure 9 presents the decoding performance of FEfficientMPT under various fine-tuning settings for unseen codes. The pretrained FEfficientMPT model was initially trained on four different LDPC codes–$(64, 32)$, $(121, 60)$, $(121, 70)$, and $(121, 80)$–for 4000 epochs.

For the $(204, 102)$ MacKay LDPC code, only a few epochs of fine-tuning are sufficient to achieve superior decoding performance. In contrast, for the $(1920, 1600)$ WiMAC LDPC code, FEfficientMPT initially struggles with decoding unseen codes. However, its performance gradually improves with fine-tuning.

These results demonstrate that leveraging a pretrained foundation model trained on short codes remains effective for decoding long codes when fine-tuning technique is applied.

# F COMPARISON OF TRAINED AND ORIGINAL PCM

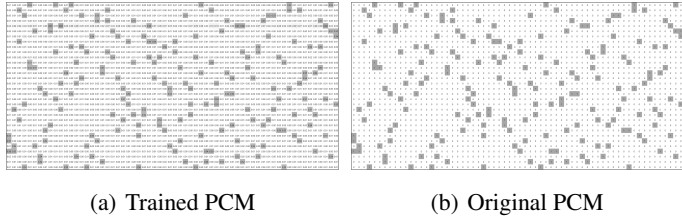

(a) Trained PCM      (b) Original PCM

Figure 10: Comparison of the trained PCM and the original PCM. The larger its value, the darker the tone of each element.

Instead of utilizing the PCM in the EfficientMPT architecture, we replace the PCM $\mathbf{H}$ with a trainable matrix for $(64, 32)$ LDPC code. After the training, we obtain the trained PCM as shown in Figure 10(a), whose structure closely related to the original PCM in Figure 10(b). This observation further reinforces the justification of incorporating the PCM in EfficientMPT.

# G COMPARISON WITH BM DECODER

We compare Berlekamp-Massey (BM) decoder with transformer-based decoders. The table shows a performance comparison between the BM decoder, ECCT, CrossMPT, and EfficientMPT. The results demonstrate that transformer-based decoders outperform the BM decoder and EfficientMPT shows comparable decoding performance with CrossMPT. Additionally, thanks to the novel EEC

attention module, EfficientMPT achieves this powerful decoding performance with significantly reduced computational complexity.

The results in this table highlight a key advantage of our approach: EfficientMPT not only serves as a universal decoder capable of handling various code classes, but also surpasses the performance of code-specific classical decoders like the BM algorithm.

Table 5: BER comparison between BM, ECCT, CrossMPT, and Proposed methods for BCH codes

| Code (n,k) | SNR | BM | ECCT | CrossMPT | EfficientMPT |
|---|---|---|---|---|---|
| | 4 dB | 1.16e-2 | 1.68e-3 | 9.26e-4 | 9.11e-4 |
| (31,16) BCH | 5 dB | 3.14e-3 | 2.51e-4 | 9.63e-5 | 8.44e-5 |
| | 6 dB | 5.49e-4 | 2.35e-5 | 3.79e-6 | 3.58e-6 |
| | 4 dB | 6.66e-3 | 7.75e-3 | 6.53e-3 | 7.44e-3 |
| (63,36) BCH | 5 dB | 9.17e-4 | 1.29e-3 | 9.98e-4 | 1.31e-3 |
| | 6 dB | 5.91e-5 | 1.12e-4 | 8.49e-5 | 9.85e-5 |
| | 4 dB | 7.80e-3 | 3.70e-3 | 2.74e-3 | 3.26e-3 |
| (63,45) BCH | 5 dB | 1.46e-3 | 4.14e-4 | 2.74e-4 | 3.20e-4 |
| | 6 dB | 1.42e-4 | 1.79e-5 | 9.01e-6 | 1.50e-5 |

## H    COMPARISON IN RAYLEIGH FADING CHANNEL

We evaluate the performance of ECCT, CrossMPT, and EfficientMPT on a Rayleigh fading channel. In previous papers, ECCT and CrossMPT are known to be robust to the non-Gaussian channels, such as Rayleigh fading channel. To compare with previous two works, we use the same fading channel as in ECCT and CrossMPT. The received codeword is given as $y = hx + z$, where $h$ is an $n$-dimensional i.i.d. Rayleigh distributed vector with a scale parameter $\alpha = 1$ and $z \sim N(0, \sigma^2)$. The following results demonstrate the BER performance of ECCT, CrossMPT, and EfficientMPT:

Table 6: BER comparison between BCH $(31, 16)$ and LDPC $(121, 70)$ codes

| Codes | BCH (31,16) | | | LDPC (121,70) | | |
|---|---|---|---|---|---|---|
| Method | 4 dB | 5 dB | 6 dB | 4 dB | 5 dB | 6 dB |
| ECCT | 5.61e-3 | 2.38e-3 | 9.85e-4 | 2.01e-2 | 6.93e-3 | 1.82e-3 |
| CrossMPT | 3.95e-3 | 1.43e-3 | 4.95e-4 | 1.42e-2 | 3.98e-3 | 8.17e-4 |
| EfficientMPT | 4.29e-3 | 1.65e-3 | 5.06e-4 | 1.45e-2 | 4.13e-3 | 8.42e-4 |

As the results demonstrate, EfficientMPT also maintains robust decoding performance even on this non-AWGN channel.

## I    PIPELINING OF EFFICIENTMPT

Although EfficientMPT may appear to be limited in throughput and latency due to its serial attention blocks, a pipelining strategy—commonly employed in various ECC decoders–can be applied to significantly enhance its decoding throughput (as illustrated in Figure 13, Appendix K in (Park et al., 2025)). By unrolling the two EfficientMPT blocks within each layer for parallel hardware implementation, the model can process two consecutive codewords simultaneously. Specifically, while the second EfficientMPT block (responsible for syndrome updates) operates on the first codeword, the first block (responsible for magnitude updates) can begin decoding the next codeword. This pipelining approach allows EfficientMPT to maintain high and competitive throughput, even in fully parallel processing environments.

## J    COMPARISON WITH NEURAL-BP BASELINE

Figure 11 compares the BER performance of EfficientMPT with a Neural-BP (NBP) decoder, using the $(576, 432)$ WiMAX and $(648, 540)$ IEEE 802.11n LDPC codes. The NBP decoder is evaluated with a maximum of 20 iterations. As the results demonstrate, our proposed EfficientMPT consistently outperforms the Neural-BP decoder across various signal-to-noise ratios for both LDPC codes.

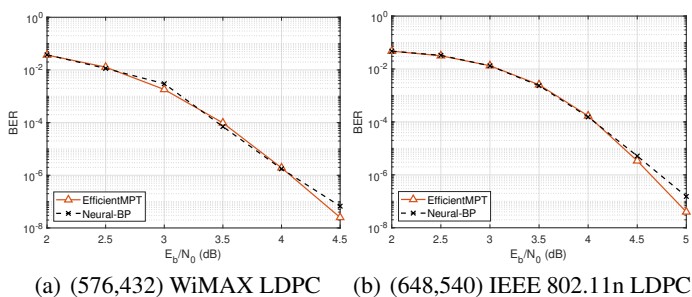

(a) (576,432) WiMAX LDPC        (b) (648,540) IEEE 802.11n LDPC

Figure 11: Decoding performance of (a) $(576, 432)$ WiMAX LDPC code and (b) $(648, 540)$ IEEE 802.11n LDPC code for the proposed method and NBP decoders.

## K    PERFORMANCE ON 5G NR LDPC CODES

Figure 12 compares the BER performance of EfficientMPT, CrossMPT, and BP decoders for Base Graph 1 $(130, 110)$ 5G NR LDPC code. EfficientMPT achieves a BER performance that is nearly identical to CrossMPT. Compared to the BP decoder with a maximum of 20 and 50 iterations, the results show that both CrossMPT and EfficientMPT outperform the BP decoder. In terms of complexity, traditional BP decoders generally exhibit lower complexity than neural network-based models. However, EfficientMPT achieves linear computational complexity, marking a substantial reduction compared to the quadratic complexity of conventional transformer-based decoders (e.g., ECCT, CrossMPT). This efficiency gain of EfficientMPT represents a major contribution towards making transformer-based decoders practical for real-world applications.

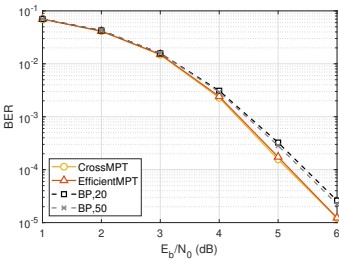

Figure 12: Decoding performance of Base Graph 1 $(130, 110)$ 5G NR LDPC code.

