# OpenReview forum: "Efficient Message-Passing Transformer for Error Correcting Codes"
_ICLR.cc/2026/Conference — ICLR 2026 Poster_

### Official Review · Reviewer_GA3B · 2025-10-16

**Soundness:** 3
**Presentation:** 2
**Contribution:** 2
**Rating:** 4
**Confidence:** 3

**Summary:**

The paper proposes an efficient variant of a transformer-based ECC decoder. The main difference from the previous methods is an efficient attention module, which is based on query and key components without a value component.

**Strengths:**

The paper conducted extensive experiments and shows improvement in memory size and computational complexity compared to previous works.

**Weaknesses:**

The contribution of the present paper, in comparison with the recent works on Transformer-based ECC decoding by Choukroun and Park, seems incremental, as it largely reproduces the same elegant algorithm with  a minor technical adjustment.

The paper is not clearly written. It is focused on the many low-level technical details and doesn't explain the intuition and motivation
of the proposed method.

**Questions:**

Can the proposed attention module architecture be useful to transformer applications other than ECC?

In line 181 it is said that standard methods have complexity o(n^2).  What is the complexity of the proposed method?

---

> ### Author Response · Authors · 2025-11-23
>
> ## Motivation and Novelty of EfficientMPT
>
> While Transformer-based decoders have achieved state-of-the-art performance on short codes, their practical application is severely constrained by the high computational complexity of the attention mechanism. Specifically, the quadratic complexity $\mathcal{O}(n^2)$ with code length $n$ of standard attention results in excessive memory usage and computational cost, making it infeasible to scale these models to longer codes.
>
> In previous transformer-based decoders, such as ECCT and CrossMPT, the parity-check matrix (PCM) is used to mask the attention map, which is obtained through matrix multiplication. This represents an indirect incorporation of the PCM (or code structure information). In contrast, our EEC attention directly integrates the PCM into the attention operation itself to transform between the magnitude domain and the syndrome domain. This represents a fundamental difference in how the PCM is utilized. Moreover, our approach eliminates the need to compute the attention map via costly matrix multiplications, which can hinder the scalability of the model for large code lengths $n$. Our model is scalable to long code lengths, which we believe represents a significant breakthrough. Thanks to this improvement, in Fig. 8, we are able to simulate long codes such as the (1824,1520) and (2304,1920) LDPC codes—code lengths that conventional transformer-based decoders cannot handle.
>
> Furthermore, the effectiveness of sparse attention techniques is limited to cases where the parity-check matrix is sparse. However, a transformer-based decoder aims to be a universal decoder, and ultimately a foundation decoder, which requires it to decode a wide variety of codes. This includes many important code families, such as BCH and polar codes, that utilize dense parity-check matrices. In such scenarios, our proposed EfficientMPT holds a significant advantage, as its efficiency is independent of the matrix structure.
>
> To summarize, EfficientMPT has three novel technical contributions:
>
> - **Global Query & Value-Free Attention:** We propose an attention mechanism using a “global query vector” that encapsulates all contextual information, simplifying the architecture by eliminating the value projection.
>
> - **Efficient Vector-Based Operation:** We replace expensive matrix multiplications with lightweight, vector-based operations, fundamentally reducing FLOPs and memory usage.
>
> - **Foundation Model for ECC Decoding:** We present a position and length-invariant architecture that serves as a foundation decoder, which also can be generalized to unseen codes via fine-tuning.
>
> ## Broader Applicability of EEC Attention
>
> While our experiments focus on error-correction tasks, we believe the architectural contributions of our model extend well beyond this domain—specifically in the context of lightweight cross-attention, which involves updating one embedding with information from another.
>
> While significant research has been devoted to designing efficient self-attention mechanisms, lightweight cross-attention remains relatively underexplored. Our work addresses this gap by introducing several key architectural innovations:
>
> 1. **Global query vector**: We compress the entire input into a single contextual representation, enabling attention without the computational overhead of a full matrix-matrix attention map.
> 2. **Value-free update mechanism**: Our method computes the update directly from the query-key interaction, eliminating the need for value projections and simplifying the attention structure.
> 3. **Additive update strategy**: Instead of relying on the traditional attention-weighted sum, we use a simple additive update, demonstrating that effective information fusion can be achieved at much lower computational cost.
>
> These design choices are not specific to error correction and can be broadly applied to tasks requiring efficient fusion of distinct representations—for example, in multimodal learning, encoder-decoder architectures, or memory-based retrieval systems.
>
> In conclusion, although ECC serves as the primary application domain in our work, we believe the underlying architectural principles are generalizable. We hope this work encourages further exploration of lightweight transformer components across a wide range of tasks and domains.
>
> ## Computational complexity of EfficientMPT
>
> As mentioned in the manuscript, the computational complexity of EfficientMPT is $\mathcal{O}(n)$, in contrast to the quadratic $\mathcal{O}(n^2)$ of standard methods. This is achieved because the EEC attention module replaces expensive matrix multiplications with linear operations, such as row-wise summation and broadcasted element-wise multiplication. While the module involves multiplication with the PCM, the use of sparse matrix multiplication for codes such as LDPC codes ensures that this operation remains efficient, preserving the overall linear complexity of the model.

---

### Official Review · Reviewer_Cs5s · 2025-10-25

**Soundness:** 4
**Presentation:** 4
**Contribution:** 4
**Rating:** 8
**Confidence:** 5

**Summary:**

This paper proposes an improved transformer-based decoder for error correction codes. The key novelty is the replacement of matrix multiplications in the attention mechanism with a vector-wise operations using a global query vector. EfficientMPT maintains decoding performance comparable to CrossMPT, while significantly reducing the parameter-count, FLOPs, and GPU usage. This is especially evident at large blocklengths - the efficiency improvements allow training at these lengths, without super-large memory usage.

**Strengths:**

1. EEC attention is well-motivated, and is a natural extension of the ideas in CrossMPT: the global query vector effectively condenses magnitude vector into one vector, rather than relying on a query matrix. The resulting updates resembles message-passing decoding between syndrome info and magnitude info, with simple updates using information from the other modality.

2. The empirical results are very strong. While maintaining the same performance as CrossMPT, efficientMPT achieves significant reductions in memory and computational complexity with a simple architectural change.

3. The main practical contribution is that it allows scaling transformer-based decoding to long blocklengths, which was infeasible via previous methods. FLOPs scale linearly with n, unlike other methods (quadratic dependence?)

4. EfficientMPT can act as a foundation model - unseen codes can be handled via lightweight finetuning.

5. I like the experiment in Figure 7, supports the understanding that a very string inductive-bias based on the PCM has been imposed on the attention mechanism.

6. The evaluation is thorough - compares with good decoders for canonical codes (SCL for polar rather than BP, etc)

**Weaknesses:**

No major weaknesses.
While transformer-based decoders (including this paper) are still sub-optimal to classical codes/decoders used in practice, this paper improves efficiency of ECCT - which is a major step in the right direction.

**Questions:**

1. How does training from scratch compare to finetuning the foundation model? Specifically, can you add FEfficient-MPT results for LDPC(1824,1520) in Figure 8a? (and/or train-from-scratch in 6c)

2. The BER performance for 5G-LDPC codes is missing. Can you please add this for completeness. I'd expect a similar performance as CrossMPT? What is the gap in performance/complexity to the decoders currently employed in 5GNR?

---

> ### Author Response · Authors · 2025-11-23
> **Authors' comments**
>
> ## Additional FEfficientMPT results for (1824,1520) LDPC code
>
> - **(1824,1520) LDPC code: https://ibb.co/LD7kZBSy**
>
> | Methods | 2 dB | 2.5 dB | 3 dB | 3.5 dB | 4 dB | 4.5 dB |
> | --- | --- | --- | --- | --- | --- | --- |
> | EfficientMPT | 4.57E-02 | 3.02E-02 | 9.56E-03 | 4.85E-04 | 2.67E-06 | 2.58E-08 |
> | BP, L=20 | 4.45E-02 | 2.83E-02 | 7.60E-03 | 2.75E-04 | 4.37E-06 | 2.67E-07 |
> | BP, L=50 | 4.44E-02 | 2.85E-02 | 7.19E-03 | 1.55E-04 | 2.04E-06 | 5.71E-08 |
> | Fefficient-0 | 4.59E-02 | 3.08E-02 | 1.18E-02 | 1.15E-03 | 2.70E-05 | 7.58E-07 |
> | Fefficient-10 | 4.58E-02 | 3.05E-02 | 1.05E-02 | 6.91E-04 | 7.56E-06 | 1.10E-07 |
> | Fefficient-100 | 4.59E-02 | 3.03E-02 | 9.88E-03 | 5.66E-04 | 4.31E-06 | 3.29E-08 |
> | Fefficient-200 | 4.57E-02 | 3.02E-02 | 9.77E-03 | 5.26E-04 | 3.99E-06 | 2.38E-08 |
>
> According to the reviewer’s comment, we conduct an additional simulation to obtain the fine-tuning performance on the (1824, 1520) LDPC code. We added the results of FEfficientMPT-10 and FEfficientMPT-200 to the existing comparison involving the scratch-trained EfficientMPT and BP decoders (originally shown in Figure 8a).
> For clarity, the number after 'BP-' and 'FEfficientMPT-’ denote the maximum number of iterations and the number of fine-tuning epochs, respectively.
> The results demonstrate that with only 10 epochs of fine-tuning (FEfficientMPT-10), the model already achieves performance comparable to the BP decoder. Furthermore, when fine-tuned for 200 epochs (FEfficientMPT-200), the performance closely matches that of the EfficientMPT trained from scratch and surpasses the BP decoder.
>
> ## Performance on 5G NR LDPC Codes
>
> - **BG1 (130,110) 5G NR LDPC code: https://ibb.co/GK81TYy**
>
> | Method | 1 dB | 2 dB | 3 dB | 4 dB | 5 dB | 6 dB |
> | --- | --- | --- | --- | --- | --- | --- |
> | CrossMPT | 7.01E-02 | 4.13E-02 | 1.48E-02 | 2.28E-03 | 1.56E-04 | 1.21E-05 |
> | EfficientMPT | 6.99E-02 | 4.14E-02 | 1.51E-02 | 2.42E-03 | 1.74E-04 | 1.23E-05 |
> | BP, L=20 | 7.03E-02 | 4.27E-02 | 1.58E-02 | 3.10E-03 | 3.25E-04 | 2.61E-05 |
> | BP, L=50 | 7.03E-02 | 4.26E-02 | 1.58E-02 | 2.93E-03 | 2.83E-04 | 2.23E-05 |
>
> We appreciate the reviewer’s suggestion to include 5G NR LDPC code results to ensure completeness. Due to the limited time available during the rebuttal phase, we conducted experiments specifically on the Base Graph 1 (130, 110) 5G NR LDPC code.
> As illustrated in the figure above, EfficientMPT achieves a BER performance that is nearly identical to CrossMPT. Compared to the BP decoder with a maximum of 20 and 50 iterations, the results show that both CrossMPT and EfficientMPT outperform the BP decoder.
> In terms of complexity, we acknowledge that traditional BP decoders generally exhibit lower complexity than neural network-based models. However, EfficientMPT achieves linear computational complexity, marking a substantial reduction compared to the quadratic complexity of conventional transformer-based decoders (e.g., ECCT, CrossMPT). This efficiency gain of EfficientMPT represents a major contribution towards making transformer-based decoders practical for real-world applications.

---

### Official Review · Reviewer_2sfM · 2025-10-29

**Soundness:** 4
**Presentation:** 4
**Contribution:** 3
**Rating:** 6
**Confidence:** 5

**Summary:**

The paper proposed a transformer based decoder for error correcting codes. The proposed decoder use a lightweight attention module that replace the costly attention mechanism, in the Masked self attention module and masked cross-attention module.
The proposed EEC decoder run over the magnitude and syndrome separately with the proposed simpler attention module.

They get very good results that reduce the number of parameters, memory usage, and FLOPs, without compromising the decoding performance achieved by CrossMPT.

**Strengths:**

1. The proposed decoder is very efficient with respect to previous methods in terms of memory usage and FLOPs

2. The proposed EfficientMPT decoder get competitive results compare to previous methods such as ECCT and CrossMPT

3. The EfficientMPT algorithm can also serve as foundation model, a single model can be generalizes to unseen codes with fine-tuning, and get better results than BP on long LDPC codes.

4. The proposed lightweight attention in the decoder using a global query vector and embedding the parity-check matrix, yields a simpler decoder.

**Weaknesses:**

1. The proposed algorithm evaluate only on simple AWGN channel, and not on non-Gaussian channels such as Rayleigh channel

2. The proposed foundation needs fine-tuning in order to operate well on larger codes.

3. For Polar codes, the classic SCL decoder still get better results than the proposed EfficientMPT decoder,

**Questions:**

1. Can you check the results of the proposed decoder on non-AWGN channel?

2. Can you suggest ways to reduce the gap to the SCL results? maybe some changes in the architectural or at loss level?

3. Please run the simulation on all codes at it appears in the paper of [Choukroun & Wolf (2022)], for example in Table 1 you only test two BCH code, instead of 4 codes and larger ones

---

> ### Author Response · Authors · 2025-11-23
> **Authors' comments**
>
> ## Comparison in Rayleigh Fading Channel
>
> We have included the comparison under the Rayleigh fading channel in **Appendix I, Table 6** of the submitted manuscript. Due to page limits, we were unable to include the table in the main body of the manuscript. .
>
> We evaluate the performance of ECCT, CrossMPT, and EfficientMPT on a Rayleigh fading channel. In previous papers, ECCT and CrossMPT are known to be robust to the non-Gaussian channels, such as the Rayleigh fading channel. To compare with previous two works, we use the same fading channel setting as in ECCT and CrossMPT. The received codeword is given as $y = hx + z$, where $h$ is an $n$-dimensional i.i.d. Rayleigh distributed vector with a scale parameter $\alpha=1$ and $z\sim N(0,\sigma^2)$. The following results demonstrate the BER performance of ECCT, CrossMPT, and EfficientMPT:
>
> | Codes | BCH   (31,16) |  |  | LDPC   (121,70) |  |  |
> | --- | --- | --- | --- | --- | --- | --- |
> | Method | 4 dB | 5 dB | 6 dB | 4 dB | 5 dB | 6 dB |
> | ECCT | 5.61.E-03 | 2.38.E-03 | 9.85.E-04 | 2.01.E-02 | 6.93.E-03 | 1.82.E-03 |
> | CrossMPT | 3.95.E-03 | 1.43.E-03 | 4.95.E-04 | 1.42.E-02 | 3.98.E-03 | 8.17.E-04 |
> | EfficientMPT | 4.29.E-03 | 1.65.E-03 | 5.06.E-04 | 1.45.E-02 | 4.13.E-03 | 8.42.E-04 |
>
> As the results demonstrate, EfficientMPT also maintains robust decoding performance even on this non-AWGN channel.
>
> ## Comparison with SCL Decoders for Polar Codes
>
> We have included the comparison with SCL decoders in **Appendix D, Table 4** in the submitted manuscript. Due to page limits, we were unable to include the table in the main text.
>
> As observed, Transformer-based decoders (including ECCT, CrossMPT, and EfficientMPT) generally do not yet outperform state-of-the-art SCL decoders for polar codes. This performance gap is a fundamental challenge for the current field of neural error correction, as Transformer-based decoders are still in the early stages of development compared to the highly mature SCL algorithm.
>
> However, while the SCL decoder is limited to polar codes, transformer-based decoders are designed as universal and foundation decoders capable of decoding various code families (e.g., BCH, LDPC, and Polar) with a single architecture. Despite the current gap in polar code performance, their flexibility and potential for improvement make them a promising direction for future research.
>
> ## Future Directions to Bridge the Gap with SCL
>
> We fully agree that bridging the performance gap with SCL decoder is an important challenge in transformer-based decoder. We concur with your view that advancements are necessary at both the architectural level and training strategy to reduce the gap to the SCL results. In particular, we believe that incorporating the unique, recursive properties of polar codes into the design will be essential.
>
> ## Performance on Additional Codes
>
> We have already conducted simulations on the full range of codes, including the additional BCH codes and larger sizes referenced in [Choukroun & Wolf (2022)], and these results are presented in **Appendix B, Table 2** in the submitted manuscript. Due to page limits, we were unable to include the table in the main body of the paper. However, the results in the Appendix confirm that, consistent with the findings in the main text, EfficientMPT achieves decoding performance comparable to CrossMPT across these various code classes.
>
> ## Final Remark
> Most of the concerns raised by the reviewer have already been addressed in the submitted manuscript, and we hope that this clarification will be taken into account in the final evaluation.

---

> > ### Comment · Reviewer_2sfM · 2025-11-26
> >
> > Thank you for addressing my concerns and for your valuable answer.

---

### Official Review · Reviewer_v8pc · 2025-11-01

**Soundness:** 3
**Presentation:** 3
**Contribution:** 3
**Rating:** 6
**Confidence:** 4

**Summary:**

This work introduces EfficientMPT, a transformer-based decoder for error-correcting codes that targets computational and memory bottlenecks in existing transformer-based decoders. The method introduces an efficient attention mechanism, replacing standard matrix multiplications with vector-based element-wise operations and integrating the parity-check matrix (PCM) directly into the attention module. EfficientMPT is designed to be position- and length-invariant, enabling it to serve as a foundation model for ECC decoding. Experimental results show significant reductions in GPU memory, FLOPs, and parameter count, while maintaining bit error rate (BER) performance comparable to existing transformer-based decoders.

**Strengths:**

- Effective handling of longer codes, overcoming a key constraint of earlier transformer-based ECC decoders.
- Position and length invariance, supporting its use as a foundation model.
- Thorough assessment across multiple code types.

**Weaknesses:**

- Limited interpretability analysis; it remains unclear how decoding decisions are made compared to traditional belief propagation.
- Missing comparisons with neural Tanner graph-based decoders.
- The analysis of polar codes would be strengthened by a more prominent and direct comparison with SCL decoders, particularly systematic SCL decoders when evaluating BER.
Further Suggestions
- Please verify the dimensions of magnitude and syndrome embedding matrices after multiplication by H and Hᵀ in Figure 2(c) (green multi-head patches).
- The sentence “The lifting process generates LDPC codes of size (52×Z,10×Z)” appears to describe the PCM dimensions rather than the code parameters (n, k). Please clarify.

**Questions:**

- The sentence “The magnitude embedding is then added to the resized syndrome embedding, which is resized from (n−k)×d to n×d by multiplying the PCM H” is unclear upon first reading—specifically, whether H is in binary or BPSK form. While later sections resolve this, earlier clarification would be helpful.
- Is any sparsity-inducing regularization applied to the trainable PCM in Figure 7(b)?

---

> ### Author Response · Authors · 2025-11-23
> **Authors' comments**
>
> ## Visualization of Syndrome Embedding $\mathbf{S}$ and EEC Attention Output $\Delta \mathbf{S}$
>
> By visualizing the syndrome embedding $\mathbf{S}$ and the update term $\Delta \mathbf{S}$ in this experiment, we demonstrate the internal mechanism of how EfficientMPT corrects errors.
>
> To investigate this, we intentionally introduce a single-bit error at the first bit position of the (64, 32) LDPC code and analyze the resulting feature maps.
>
> - **Fig. R1(a): Syndrome embedding $\mathbf{S}$ - [https://ibb.co/JFHGJm37](https://ibb.co/whJ2Yx6F)**
> - **Fig. R1(b): EEC output embedding $\Delta \mathbf{S}$ - [https://ibb.co/LdLSfJTn](https://ibb.co/0RN04WwV)**
>
> Specifically, the above Figs. R1 (a) and (b) visualize (1) the syndrome embedding $\mathbf{S}$ and (2) the EEC attention output $\Delta \mathbf{S}$, which represents the syndrome update term—both averaged across the feature dimension using absolute values.
> As shown in the figure, the initial syndrome embedding (Layer 1) has a relatively uniform distribution across positions. In contrast, the activations in $\Delta \mathbf{S}$ at the initial layer (Layer 1) are distinctively concentrated; the most prominent positions correspond exactly to the indices of the check nodes connected to the erroneous variable node—specifically, indices 25, 26, and 28. Since the syndrome embedding is updated via $\mathbf{S} \leftarrow \mathbf{S} + \Delta \mathbf{S}$, the update values $\Delta \mathbf{S}$ are directly accumulated into $\mathbf{S}$. Consequently, we observe that the updated syndrome embedding $\mathbf{S}$ also exhibits distinctively high activations at these specific check node indices. This indicates that the model initially focuses heavily on the syndrome embeddings connected to the corrupted bit. However, in deeper layers, the distinction between the connected check nodes and the others diminishes. Consequently, these visualizations effectively demonstrate the mechanism how EfficientMPT corrects errors.
>
> ## Comparison with Neural-BP Baseline
>
> Following your suggestion, we conducted additional experiments comparing EfficientMPT with a Neural-BP (NBP) decoder, using the (576, 432) WiMAX and (648, 540) IEEE 802.11n LDPC codes in Fig. 4. We compared our model against NBP with a maximum of 20 iterations. The results are presented in the figure below:
>
> | (576,432) LDPC | 2 dB | 2.5 dB | 3 dB | 3.5 dB | 4 dB | 4.5 dB |
> | --- | --- | --- | --- | --- | --- | --- |
> | EfficientMPT | 3.68.E-02 | 1.28.E-02 | 1.81.E-03 | 9.73.E-05 | 1.92.E-06 | 2.50.E-08 |
> | Neural-BP | 3.76.E-02 | 1.14.E-02 | 3.00.E-03 | 7.04.E-05 | 1.76.E-06 | 6.72.E-08 |
>
> | (648,540) LDPC | 2 dB | 2.5 dB | 3 dB | 3.5 dB | 4 dB | 4.5 dB | 5 dB |
> | --- | --- | --- | --- | --- | --- | --- | --- |
> | EfficientMPT | 4.72.E-02 | 3.16.E-02 | 1.33.E-02 | 2.54.E-03 | 1.69.E-04 | 3.39.E-06 | 3.98.E-08 |
> | Neural-BP | 4.71.E-02 | 3.28.E-02 | 1.32.E-02 | 2.35.E-03 | 1.54.E-04 | 5.15.E-06 | 1.53.E-07 |
>
> As the new results demonstrate, our proposed EfficientMPT consistently outperforms the Neural-BP decoder across various signal-to-noise ratios for both LDPC codes. We thank you for this valuable suggestion, as this new comparison more clearly highlights the performance gains of our method.
>
> ## Comparison with SCL Decoders for Polar Codes
>
> We have included the comparison with SCL decoders in **Appendix D (Table 4)** due to the page limit of the main text.
>
> As shown, transformer-based decoders (including ECCT, CrossMPT, and EfficientMPT) generally do not yet outperform state-of-the-art SCL decoders for polar codes. However, EfficientMPT achieves substantially better performance than the original transformer-based decoder, ECCT, while maintaining significantly lower complexity.
> Although the field of transformer-based decoders is still in an early stage compared to the highly mature SCL decoders, the performance gap has been rapidly decreasing. Notably, EfficientMPT surpasses the conventional SC decoder (i.e., SCL with $L=1$) for polar codes, even though EfficientMPT is not specifically designed for polar codes but is instead applied universally across various code families (BCH, LDPC, and Polar). Despite the current gap in polar code performance, their flexibility and potential for further improvement make transformer-based decoders a promising direction for future research.

---

> ### Author Response · Authors · 2025-11-23
> **Authors' comments**
>
> ## Clarification on Embedding Dimensions in Figure 2(c)
>
> Regarding the dimensions of the embeddings in Figure 2(c), the total embedding dimension $d$ is divided into $h$ heads, where each head has a dimension of $d_h$ (i.e., $d_h=d/h$).
> The magnitude embedding $\mathbf{M}$ initially has dimensions of $n \times d$. After passing through the Linear layer, it retains the same dimensions $n \times d$. Subsequently, multiplying this by the parity-check matrix $\mathbf{H}$ results in a matrix of size $(n-k) \times d$. This matrix is then split into $h$ heads. Consequently, the ‘green multi-head patches’ shown in the figure correspond to $h$ separate matrices, each with dimensions of $(n-k) \times d_h$. We will revise the figure in our final manuscript to ensure clarity.
>
> ## Clarification on LDPC Code Dimensions
>
> "We appreciate the reviewer for pointing out the potential ambiguity. We clarify that the notation $(52 \times Z, 10 \times Z)$ was intended to represent the code parameters $(n, k)$. To strictly distinguish this from the PCM dimensions and ensure clarity, we will revise the sentence in our final manuscript as follows:
>
> • **Original:** 'The lifting process generates LDPC codes of size $(52 \times Z, 10 \times Z)$.'
>
> • **Revised:** 'The lifting process generates LDPC codes with code length $n = 52 \times Z$ and dimension $k = 10 \times Z$.'
>
>
> ## Clarification on the Binary Form of PCM H for Resizing
>
> We clarify that the operation utilizes the **binary** parity-check matrix. To explicitly state this and resolve any confusion upon first reading, we will revise the sentence in our final manuscript as follows:
>
>
> • **Original:** 'The magnitude embedding is then added to the resized syndrome embedding, which is resized from $(n-k)\times d$ to $n\times d$ by multiplying the PCM $H$.'
>
>
> • **Revised:** 'The magnitude embedding is then added to the resized syndrome embedding, which is resized from $(n-k)\times d$ to $n\times d$ by multiplying the transpose of the **binary** PCM $H$.'
>
> ## Training PCM in EfficientMPT
>
> When learning PCM with trainable matrix, **we did not apply any regularization**. The trainable matrix resembles the original PCM naturally after the training.

---

### Meta-Review · Area_Chair_vzAi · 2026-01-09

**Summary:**

- EEC attention is well-motivated, and is a natural extension of the ideas in CrossMPT: the global query vector effectively condenses magnitude vector into one vector, rather than relying on a query matrix.
- The main practical contribution is that it allows scaling transformer-based decoding to long blocklengths, which was infeasible via previous methods.
- EfficientMPT can act as a foundation model - unseen codes can be handled via lightweight finetuning.

**Reviewer Concerns:**

These concerns have been properly addressed.

- The contribution of the present paper, in comparison with the recent works on Transformer-based ECC decoding by Choukroun and Park, seems incremental, as it largely reproduces the same elegant algorithm with a minor technical adjustment.

- The paper is focused on the many low-level technical details and doesn't explain the intuition and motivation of the proposed method.

**Reviewer Scores:**

I believe Reviewer GA3B would have increased their score.

---

### Decision · Program_Chairs · 2026-01-26

Accept (Poster)